# The Change P82L in the Rift Valley Fever Virus NSs Protein Confers Attenuation in Mice

**DOI:** 10.3390/v13040542

**Published:** 2021-03-24

**Authors:** Belén Borrego, Sandra Moreno, Nuria de la Losa, Friedemann Weber, Alejandro Brun

**Affiliations:** 1Centro de Investigación en Sanidad Animal, INIA-CISA, 28130 Valdeolmos, Spain; borrego@inia.es (B.B.); moreno.sandra@inia.es (S.M.); dllosa@inia.es (N.d.l.L.); 2Institut für Virologie, Justus-Liebig-Universität, D-35392 Giessen, Germany; friedemann.weber@vetmed.uni-giessen.de

**Keywords:** Rift Valley fever virus, non-structural NSs protein, interferon antagonist, nuclear filaments, PXXP motifs

## Abstract

Rift Valley fever virus (RVFV) is a mosquito-borne bunyavirus that causes an important disease in ruminants, with great economic losses. The infection can be also transmitted to humans; therefore, it is considered a major threat to both human and animal health. In a previous work, we described a novel RVFV variant selected in cell culture in the presence of the antiviral agent favipiravir that was highly attenuated in vivo. This variant displayed 24 amino acid substitutions in different viral proteins when compared to its parental viral strain, two of them located in the NSs protein that is known to be the major virulence factor of RVFV. By means of a reverse genetics system, in this work we have analyzed the effect that one of these substitutions, P82L, has in viral attenuation in vivo. Rescued viruses carrying this single amino acid change were clearly attenuated in BALB/c mice while their growth in an interferon (IFN)-competent cell line as well as the production of interferon beta (IFN-β) did not seem to be affected. However, the pattern of nuclear NSs accumulation was modified in cells infected with the mutant viruses. These results highlight the key role of the NSs protein in the modulation of viral infectivity.

## 1. Introduction

Rift valley fever virus (RVFV) is a mosquito-borne phlebovirus of the *Phenuiviridae* family (*O. Bunyavirales*) that causes an important disease in ruminants, mostly characterized by a high-rate of abortions, fetal malformation and death of newborn lambs, with great economic losses. The infection can be transmitted to humans through mosquito bites or when exposed to infected material, producing a usually self-limiting disease with more severe development in a low percentage of cases (reviewed in [1]). Rift Valley fever is confined to the African continent and southern parts of the Arabian Peninsula and Indian Ocean islands, but its potential for spreading to other geographical areas linked to climatic change and globalization has been widely remarked [2]. Veterinary vaccines are available in Africa, but currently there are no licensed vaccines for human use, while in Europe there is no available treatment or licensed RVF vaccine. Therefore, the development of safer and effective control strategies intended also for human use is an active field of research [3,4,5].

The RVFV genome consists of three ssRNA(-) segments of different sizes (large, medium, small). The L-segment codes for an RNA-dependent RNA polymerase (RdRp). The M segment contains five in-frame start codons alternatively used by virtue of a ribosomal “leaky scanning” mechanism for the synthesis of the envelope glycoproteins (Gn and Gc), a cytosolic accessory protein (NSm) that can be found in two isoforms of 13–14-kDa protein [6], and a 78-kDa glycoprotein (NSm-Gn) that incorporates in virus particles when produced in insect cells [7] but with unknown functions in mammal hosts. The S segment encodes in an ambisense strategy the viral 27 kDa nucleoprotein (N), and a 30kDa protein (NSs), considered the main virulence factor of the virus.

The NSs protein inhibits the host antiviral responses by multiple pathways that, either alone or combined, allow the virus to replicate efficiently. NSs interaction with several binding partners promotes the sequestration or degradation of a number of cellular proteins, thus avoiding their functions: NSs prevents the activation of the interferon (IFN)-ß promoter, promotes the degradation of double-stranded RNA-dependent protein kinase R (PKR) and blocks the assembly of transcription factor II H (TFIIH), inducing a general transcription shut off in infected cells (reviewed in [8,9]). These biological functions seem to be dependent on the nuclear localization of NSs in the infected cells, where it assembles displaying a typical filamentous pattern unique in phleboviruses. Another role related with the accumulation of superoxide in infected cells has been suggested for NSs, in association with non-nuclear compartments, such as mitochondria [10]. Several works have identified different amino acid positions or regions of the protein involved in different functions. An essential core domain spanning residues 83–248 was shown to be sufficient for filament formation [11]; other results suggest that the NSs functionality is more likely dependent on conformational integrity than on the presence of particular domains [12]. However, the understanding of the whole picture is still unclear, especially regarding the relationships between some of these functions and their combined contribution to virulence in vivo. The characterization of the biological features of NSs is a big step towards the development of effective control measures for RVF. Due to its role in providing an efficient viral replication, NSs appears to be a good target for antivirals and, in addition, some live attenuated vaccines are based on viruses lacking or carrying a non-functional NSs protein. Besides these biological functions, recent data showed that the reported nuclear NSs filamentous pattern corresponds to amyloid-like structures that could play an important role in mouse neuropathology or neurotoxicity [13].

In a previous work aimed to characterize a novel RVFV variant that was selected in cell culture in the presence of the antiviral compound favipiravir, we found that this virus, named as 40F-p8, was highly attenuated in vivo [14]. Out of the 24 amino acid substitutions found in other viral proteins when compared to the parental virulent strain, only two changes were located on the NSs protein: V52I and P82L. Since V52I is a conservative substitution and variants at this position have been reported in other RVFV strains (V52A in Madagascar strains 0212-08 and 200803162) we focused on the P82L mutation. P82 is placed within the PXXP motif 2 (positions 82 to 85), a motif reported to be involved in both NSs nuclear localization and IFN-β activation/expression, with proline residues playing a critical role [15]. In this work, using a reverse genetics system, we investigated the role of NSs P82L mutants on the viral infectivity in vivo in a mouse model of infection.

## 2. Materials and Methods

### 2.1. Cells

The cell lines used for this study were HEK293T (human embryonic kidney 293 cells, ATCC CRL-3216), Vero (African green monkey kidney cells, ATCC CCL-81) and BHK-21 (baby hamster kidney fibroblasts, ATCC CCL-10). All cell lines were grown as described [16]. Recombinant RVFV (rRVFV) previously generated (rZH548 (wild-type) and a NSs-deleted virus expressing green fluorescent protein, named as rZH548ΔNSs::GFP, [16]) were included as controls in the different assays. Infections were performed as described [14]. Assays to quantify plaque-forming units (pfu) were carried out on Vero cells in semisolid medium consisting of Dulbecco’s Modified Eagle Medium (DMEM 1X)-1% Carboxymethylcellulose (CMC, Sigma, St. Louis, MO, USA). Monolayers were fixed and stained 5 days post infection.

### 2.2. huIFN-β ELISA

The levels of human IFN-β in infected HEK293T cell supernatants were tested using the IFN-β Human ELISA Kit (PBL Assay Science, Piscataway, NJ, USA), following manufacturer’s instructions. Briefly, a standard curve in the range of 50–4000 pg/mL was constructed and used to calculate the interferon titer in each sample by plotting the mean OD value obtained for each sample, that was tested undiluted in duplicate. Blank ODs were subtracted in all cases. Samples rendering an optical density (OD) value lower than the one in wells with buffer alone were considered as negative and for graphic representation an arbitrary value of 25 was assigned. Since 50 pg/mL was the lowest concentration of huIFN-β in the standard curve, this was the limit of detection of the assay. 

### 2.3. Rescue of Recombinant Viruses

Recombinant RVF viruses were rescued by means of a reverse genetic system [16,17]. Briefly, this system is based on the transfection of HEK293T and BHK-21 cell co-cultures of a set of 5 plasmids comprising 3 plasmids providing viral genomic segments L, M and S (pHH21_RVFV_vL, pHH21_RVFV_vM and pHH21_RVFV_vS, respectively) and 2 plasmids providing the viral polymerase L and the nucleoprotein N (pI.18_RVFV_L and pI.18_RVFV_N).

To generate the plasmid carrying a mutant S segment, the desired nucleotide change C279T (numbering according to NC_014395 RVFV segment S, strain “ZH-548”) was introduced in plasmid pHH21-RVFV-vS by PCR using the Q5^®^ Site-Directed Mutagenesis Kit (NewEngland Biolabs, Ipswich, MA, USA) following manufacturer’s protocols. The primers used, designed using the NEB online design software NEBaseChanger™, were S279Fwd (5′-GCACCTCCACTAGCGAAGCCT-3′; underlined letter corresponds to the nucleotide changed) and S279Rev (5′-AACGTTTGATGCAAAGTCTCCAAGTC-3′). On days 3, 5 and 7, transfected cell supernatants were harvested and inoculated onto BHK-21 cells in order to screen for the presence of virus by cytopathic effect (CPE). For those rendering positive CPE, further two passages were performed to generate a virus stock for use in the present experiments. 

### 2.4. Animal Inoculation and Sampling

BALB/c mice (9–18-week-old male) were used for the in vivo studies. They were equally distributed into groups of 5–7 animals and inoculated intraperitoneally with 500 plaque-forming units (pfus) of the corresponding viruses. Development of disease was evaluated over 3 weeks (18 days) in terms of mortality and morbidity, as elsewhere [18] checking for weight and development of clinical signs, such as ruffled fur, ocular discharges, hunched posture and reduced activity. Blood samples were collected in Microvette^®^ tubes K3 EDTA or Serum (Sarstedt, Nümbrecht, Germany) upon submandibular puncture at 72 h after infection and tested for viral RNA by RT-qPCR [16,19] to monitor viremia, while serum samples collected from survivor mice at the end of the experiment (day 18 pi) were used in antibody assays. Mice were housed in biosafety level 3 (BSL-3) animal facilities at INIA-CISA before use. All experimental procedures involving animals were performed in accordance with EU guidelines (directive 2010/63/EU), and protocols approved by the Animal Care and Biosafety Ethics’ Committees of INIA and Comunidad de Madrid (permit codes CEEA 2014/26, CBS 2017/15, PROEX 108/15 and PROEX192/17). 

### 2.5. Sequencing and RT-qPCR Assays

RNA was extracted from supernatants of cells infected with recombinant viruses of passage 3 after rescue or from blood samples collected at day 3 pi using an RNA virus extraction kit (Speedtools, 180 Biotools BM, Madrid, Spain) as described [16]. For sequencing, amplicons corresponding to the S-segment were obtained by RT-PCR using a SuperScript IV Reverse Transcriptase and a Phusion High-Fidelity DNA polymerase (Thermofisher, Whaltham, MA, USA), as described [14]. PCR products were purified and the sequence corresponding to the NSs ORF (RNA positions 35–832) was determined by automated Sanger-sequencing. The Lasergene software suite (DNAstar, Madison, WI, USA) was used for sequencing data analysis.

To monitor viremia, a real-time RT-qPCR specific to the RVFV L-segment [19] was performed on RNA extracted from blood samples. To establish a correspondence between Cq values and plaque forming units (pfus), blood from naïve mice was spiked with 10^1^ to 10^5^ pfu of a plaque assay titrated RVFV stock and RNA was extracted for RT-qPCR as above.

### 2.6. Antibody Assays

Antibodies against the viral nucleoprotein N were detected by an in-house ELISA and RVFV neutralizing antibodies in a microneutralization assay [14]. Briefly, for the detection of antibodies against N, sera were tested in duplicate in serial 3-fold dilutions starting at 1/50, in ELISA plates adsorbed with 100 ng/well of purified recombinant Thioredoxin-N (Trx-N) fusion protein produced in *Escherichia coli* and diluted in carbonate buffer (pH 9.6). Wells were blocked with 5% skimmed milk in PBS, 0.05% Tween 20, then bound antibodies were detected with goat anti-mouse-IgG (H+L)-HRP Conjugated (Bio-Rad, Hercules, CA, USA) and TMB (Thermofisher) was used as chromogen substrate. For neutralization, sera (in quadruplicates) were 2-fold diluted from 1/10, mixed with an equal volume of infectious virus containing 100 TCID_50_ and incubated for 30 minutes at 37 °C. Then, a Vero cell suspension was added, and plates were incubated for 4 days. Monolayers were then controlled for the development of the cytopathic effect (CPE), fixed and stained. Anti-N titers are expressed as last dilution of serum (log10) giving an OD reading at 450 nm over 1.0 in ELISA; neutralization titers are expressed as the dilution of serum (log10) rendering a reduction in infectivity of 50%. 

### 2.7. Immunofluorescence

Vero cells were infected at a multiplicity of infection (MOI) of 1 and, at the time post-infection (pi) indicated, cells were fixed with 4% paraformaldehyde and subjected to indirect immunofluorescence with the anti-NSs monoclonal antibody 5C3A1B12 [20], kindly provided by Dr. Martin Eiden (Friedrich-Loeffler Institute, Riems, Germany) following procedures as described [16]. All the buffers included 0.1% saponin for permeabilization. The secondary antibody was a goat anti-mouse Alexa Fluor 488. Cell nuclei were stained with DAPI. Microscopy was performed with a Zeiss LSM880 confocal laser microscope (Gmbn, Oberkochen, Germany). 

### 2.8. Western Blot

HEK293T cells were infected at a MOI of 1 and whole cell extracts harvested at 20 hpi and lysed in Laemmli’s SDS-PAGE sample buffer (Bio-Rad). Proteins were transferred to a Whatmann nitrocellulose membrane (Merck, Darmstadt, Germany). The membrane was incubated for one hour with 5% skimmed milk in Tris-buffered saline (TBS). Upon blocking, incubation with primary antibodies diluted in 5% milk-TBS-T (TBS with 0.05% Tween-20) was performed at 4 °C overnight. The antibodies used were: anti-PKR (B-10) and anti-p62 (H10) mouse monoclonal antibodies (Santa Cruz Biotechnology, Dallas, TX, USA), anti RVFV-N mAb 2B1 [16] and mouse anti-actin antibody (Sigma), at dilutions 1/200; 1/250; 1/1000, and 1/2000, respectively. The membranes were then washed three times with TBS-T and incubated for 1 hour at room temperature with an anti-mouse IgG-HRPO conjugated antibody. The membrane was washed again three times with TBS-T before adding luminiscent substrate (ECL-GE Healthcare, Little Chalfont, Buckinhamshire, UK). Gel images were visualized using a ChemiDoc Imaging System (Bio-Rad) and were analyzed by Western blot. Loaded samples corresponded to 10^6^ cells per well. 

### 2.9. Statistical Analysis

Data analysis was performed using GraphPad Prism software (version 6.0). Differences in survival times were tested by the Log-Rank (Mantel–Cox) test. Variations in the mean viral titers were analyzed using a non-parametric one-way ANOVA test (Kruskall–Wallis test) with Dunn’s multiple comparison post hoc tests. Differences were considered statistically significant when *p* < 0.05.

## 3. Results

### 3.1. Rescue of Recombinant Rift Valley Fever Viruses Carrying the S-Segment C279T Substitution 

We planned to rescue recombinant ZH548 (rZH548) viruses carrying the amino acid substitution P82L in the NSs protein by means of our reverse genetic system [17]. This amino acid change was deduced from the nucleotide sequence of the virus 40F-p8 that displayed the change C279T in the corresponding codon (CCA in the parental virus RVFV 56/74, CTA in the selected variant; [14]). Thus, we first introduced the desired nucleotide change C279T in the plasmid corresponding to genomic S-segment, pHH21-RVFV-vS. After checking the correct sequence of the resulting plasmid, co-cultures of HEK293T and BHK-21 cells were transfected in triplicates with the whole set of plasmids constituting our reverse genetic system. At days 3, 5 and 7 post-transfection supernatants were harvested and inoculated onto BHK-21 cells in order to screen for the presence of virus by the appearance of cytopathic effect (CPE). In samples collected at days 5 and 7, post-transfection from two separate replica wells total CPE was observed at day 4 pi. Viruses were grown on BHK-21 cells for three passages, with CPE registered at 48 hpi and yields of 8.4- and 3.2 × 10^7^ pfu/mL, comparable to the wt rZH548 (2.7 × 10^7^ pfu/mL). Plaque formation on Vero cells was indistinguishable to rZH548 (not shown). The presence of the mutation was confirmed by RT-PCR amplification of genomic S-segment and further sequence analysis and no other changes were detected in the NSs gene. Viruses were generically named after the amino acid position changed (rZH548-P82L viruses). In particular, the two viruses obtained after three passages were termed 2B7 and 3VB5 and are the viral clones used for this study.

### 3.2. Analysis of Pathogenicity and Immunogenicity of Recombinant rZH548-P82L Mutant Viruses in Mice

The virulence of the recombinant rZH548-P82L mutant viruses was tested in BALB/c mice by intraperitoneal (ip) inoculation of 500 plaque-forming units (pfus) of the two rescued viruses, 2B7 and 3VB5. Rescued rZH548 (wild-type) and an NSs-deleted virus expressing green fluorescent protein, named as rZH548ΔNSs::GFP, were included as controls for virulence and attenuation, respectively (Figure 1).

In the control group inoculated with the attenuated, NSs-deleted, rZH548 virus, one mouse showed, unexpectedly, tremors and a severe weight loss (20%) one day after bleeding and was euthanized (day 4). We considered that this mouse did not recover well from the stress or damage caused by the bleeding procedure. It was later found to be virus negative by RT-qPCR and was excluded from the statistical analysis; thus a 100% survival was considered for this group. No signs of disease were observed in any animal within this group.

In mice inoculated with wt rZH548, the first signs of disease (watery eye, reduced mobility, ruffled fur) appeared at day 3 with first deaths occurring at day 4; disease evolved rapidly (hunched back, lethargy, paralysis) and at day 9 all the animals had deceased. The median survival time of this group was 6 days. In contrast, animals inoculated with the rZH548-P82L viruses remained without signs of disease the first week after inoculation, when disease signs were first observed. At the end of the experiment (day 18 pi) both mutant groups recorded significantly higher survival rates than the wt group: two out of seven mice inoculated with 3VB5 survived, with a median survival time of 14 days and four out of seven in the group inoculated with 2B7. This increased survival percentage of 2B7 with respect to 3VB5 did not reach statistical significance (*p* = 0.1964, Log-Rank Mantel–Cox test).

Viral loads at day 3 were analyzed by RT-qPCR (Figure 2A). As expected, all samples recovered from mice inoculated with rZH548 revealed high viral loads, while viral RNA in samples from rZH548ΔNSs::GFP-infected mice was below the detection level. In mice inoculated with rZH548-P82L viruses viral loads at day 3 pi were strongly reduced, with one animal also rendering undetectable RNA levels (2B7 group). A total of six RNA samples (three from each mutant group, randomly selected) extracted from day 3 blood were used for RT-PCR amplification and sequencing of the NSs ORF, confirming the presence of the mutation and no other changes.

These results of viremia correlated with the levels of seroconversion to viral proteins detected in survivor mice (Figure 2B). Anti-N antibodies, indicative of viral replication, were detected in all mice, including also those inoculated with rZH548ΔNSs::GFP virus where no viral RNA was detected. Likewise, antibodies able to neutralize RVFV infectivity in vitro were detected in all animals. Compared to those reached in the rZH548ΔNSs::GFP group, titers in both assays were slightly higher in groups inoculated with the rZH548-P82L viruses, although these differences were not statistically significant (one-way ANOVA test).

### 3.3. Cellular Localization and Pattern of NSs in Cells Infected with rZH548-P82L Mutants

Once it was proved that the change introduced in the NSs led to attenuation of RVFV in mice, we performed some in vitro assays to further characterize the phenotype of the P82L-mutant viruses. First, we tested the pattern of cellular distribution of the mutated NSs in Vero-infected cells. P82 is placed within PXXP motif 2 (positions 82 to 85), and changes in two of the four PXXP motifs present in the NSs (motif 1 at positions 29 to 32, and motif 2 at positions 82 to 85) are known to affect the nuclear filamentous arrangement of NSs, with NSs mutants remaining in the cytoplasm [15]. In order to test whether the change carried by our P82L-mutants affected this pattern, infected Vero cells were subjected to indirect immunofluorescence with the anti-NSs monoclonal antibody 5C3AB12 [20]. At 6 hpi, NSs could be detected in the cytoplasm of all infected cells in all cases, but in the nucleus the prototypical fibrillar NSs structures could only be detected after infection with wt rZH548 virus (Figure 3A, left panels). In contrast, in cells infected with the NSs-mutant viruses, this typical nuclear staining was harder to find at this early point: nuclear filaments were only detected in 2.0%–2.5% of the infected cells (4/162 for 2B7; 2/99 for 3VB5; counting on five different fields each), while in cells infected with the rZH548, this proportion reached 74.5% (111/149). When present, filaments seemed less defined. Rather, the fluorescence had a punctate pattern distributed along the cytoplasm and the cell nucleus (Figure 3A, central and right panels). At 24 h pi, filamentous structures could be detected in the nucleus in all cases, although some subtle morphologic differences were found again between filaments formed in cells infected with the wt (Figure 3B, left panels) and the NSs-mutant viruses (Figure 3B, central and right panels). While in rZH548-infected cells nuclear filaments appeared thicker and sharply defined, those in cells infected with both rZH548-P82L viruses looked more disordered and loosely aggregated and with more cytoplasmic staining.

### 3.4. Growth and IFN-β Induction of rZH548-P82L Mutants on HEK293T Cells

NSs is an antagonist of the antiviral type I interferon (IFN) system. RVFVs lacking NSs or a functional NSs are unable to counteract the IFN response, thus showing an impaired growth in interferon-competent cells [15,21,22]. As for the nuclear pattern of NSs, changes in the PXXP motifs have also been reported to affect the ability to suppress the activation of IFN-β promoter, in particular when prolines were substituted [15]. Thus, we decided to check whether the change introduced had some effect on the growth of rZH548-P82L viruses on interferon-competent HEK293T cells. Both the rZH548ΔNSs::GFP and the wt rZH548 virus were again included for a comparison as examples of IFN-sensitive or non-sensitive viruses, respectively (Figure 4). As expected, titers of the rZH548ΔNSs::GFP did not increase over time. In contrast, the growth curves of the two P82L mutant viruses were similar to the one displayed by the wt rZH548 virus, showing increasing titers along the time analyzed. In addition, these supernatants were analyzed for the presence of human IFN-β by ELISA. IFN-β was only detected in samples recovered 48 hpi from cells infected with rZH548ΔNSs::GFP. All the other samples rendered OD values corresponding to IFN levels below or close to the sensitivity threshold of the assay (50 pg/mL). Altogether, these results suggested that the change introduced in rZH548-P82L viruses did not affect the ability of NSs to block the cellular production of IFN and thus their growth in these IFN-competent cells.

### 3.5. Degradation of PKR and p62 in rZH548-P82L Infected Cells

Another pathway by which NSs blocks the host antiviral responses of infected cells is the degradation of cellular proteins such as PKR and p62, a component of transcription factor II H (TFIIH) [23,24,25]. In order to determine whether the change introduced affected this ability and could therefore modulate viral attenuation, HEK293T cells were infected and whole cell extracts analyzed by Western blot (Figure 5). In samples from cells infected with the two rZH548-P82L viruses, the expression of PKR was clearly diminished while p62 was undetectable, suggesting that the change in P82L did not impair the ability of the mutant protein to degrade the cellular proteins under study.

## 4. Discussion

In this work, we describe the rescue of recombinant ZH548 (rZH548) RVF viruses carrying a P82L mutated NSs protein and analyze the effect of this change in RVFV infectivity. This mutation was one of the 24 amino acid changes originally identified in a virus isolated under the selective pressure of a mutagenic agent that was found to be hyper-attenuated in mice [14,26]. Among the many substitutions identified in this RVFV variant, changes in the nucleotide 279 of genomic S-segment led to the substitution Proline➔Leucine in residue 82 in the NSs protein. Changes in this protein, known to be the main virulence factor for RVFV, were especially interesting to study. The substitution of proline 82 was of special interest since it lies within a PXXP motif involved in the correct nuclear localization of the protein and in the ability to suppress IFN-β promoter activation [15]. 

When viruses carrying P82L NSs were tested in vivo, viral loads were reduced, disease appeared later than in controls and survival rates were higher, confirming that this substitution led to virus attenuation in mice. A third additionally rescued virus (clone 3B5) displayed also longer survival times, supporting this observation (Appendix A). Although mortality rates between the clones tested were not different with statistical significance, there were slight differences in how infection progressed. While these differences could be due to normal variation in animal experimentation with a low number of individuals, we cannot exclude that minor subpopulations generated during virus growth in vitro or during viral replication in vivo might also contribute to modulate the observed pathogenicity. Nonetheless, the genomic background in which P82L was introduced corresponded to a strain of different lineage (Egyptian) with respect to the parental virus (South African). Therefore, this emphasizes the dominant role of this change in the phenotype observed. Surprisingly, none of the in vitro assays performed revealed a clear difference between rZH548wt and both mutant viruses: they all were able to grow in IFN-competent HEK293T cells and block the cellular production of IFN-β, retaining the ability to degrade both PKR and p62 proteins.

The only feature where we identified a difference between the wt and the mutant viruses was the pattern/kinetics of nuclear distribution of NSs in infected cells. Even though filamentous structures were observed in the nucleus in all cases, the assembly of these typical structures was somehow impaired for the mutant NSs, appearing with some delay and with a looser consistency at later times pi, suggesting some deficiency in nuclear import, aggregation capability or kinetics. It is difficult to determine whether this is an actual trait with any effect on the NSs activities, not only in vitro, but especially in vivo. Correlation between this typical pattern of RVFV NSs and in vivo virulence is controversial since it was described that nuclear filament formation is important but not sufficient for in vivo virulence [27,28]. Interestingly, recent data reported that the nuclear NSs filamentous pattern indeed corresponds to amyloid-like structures, therefore stressing the potential role of NSs in mouse neuropathology or neurotoxicity [13]. Further work is needed to assess the role of the P82L change in amyloid formation.

Our results provide some noteworthy findings for the development of live attenuated vaccines. In terms of attenuation, the substitution of residue P82L has a remarkable effect in mice, with higher survival than the wt virus. Most animals infected with the mutant viruses developed detectable viremia and strong seroconversion to RVFV, both at higher levels than mice inoculated with attenuated ∆NSs virus expressing green fluorescent protein (rZH548ΔNSs::GFP). When developing safer and more stable live-attenuated vaccines, a whole deletion provides a better approach than single amino acid substitutions, with lower chances of changes and reversion. Nonetheless, a total lack of NSs may lead to poorly immunogenic viruses, thus a different LAV strategy based on viruses keeping the NSs but including additional combinations of single attenuation changes may be preferred [29]. In this case, the P82L substitution could be included as an additional safety feature, since our results show that it does not affect the viral growth (production) or immunogenicity. Of note, the P82L mutation was found in a virus derived from a South African origin (lineage D) [30]. Here, we report the biological consequences of this change in the context of the ZH548 backbone (lineage A) stressing the relevance of amino acid residue 82 among distinct RVFV lineages. 

On the other hand, how this single change in the NSs leads to in vivo attenuation remains unknown. Except for a tenuous difference in the consistency and definition of the nuclear filaments, all other NSs features analyzed in this work known to affect virulence did not show significant differences between the virulent rZH548 and the mutant viruses. Virus growth and yield in Vero cells were equivalent and the rZH548-P82L viruses retained the ability to block IFN production in IFN-competent cells and to degrade cellular PKR and p62. Interactions of NSs with other cellular proteins related with mitochondrial or nuclear targeting or further interfering with the host antiviral response, as well as other effects influencing apoptosis or different immunological pathways, may contribute to the attenuated phenotype observed in mice. Work is in progress to determine the pathway affected by the substitution P82L studied in this work.

## Figures and Tables

**Figure 1 viruses-13-00542-f001:**
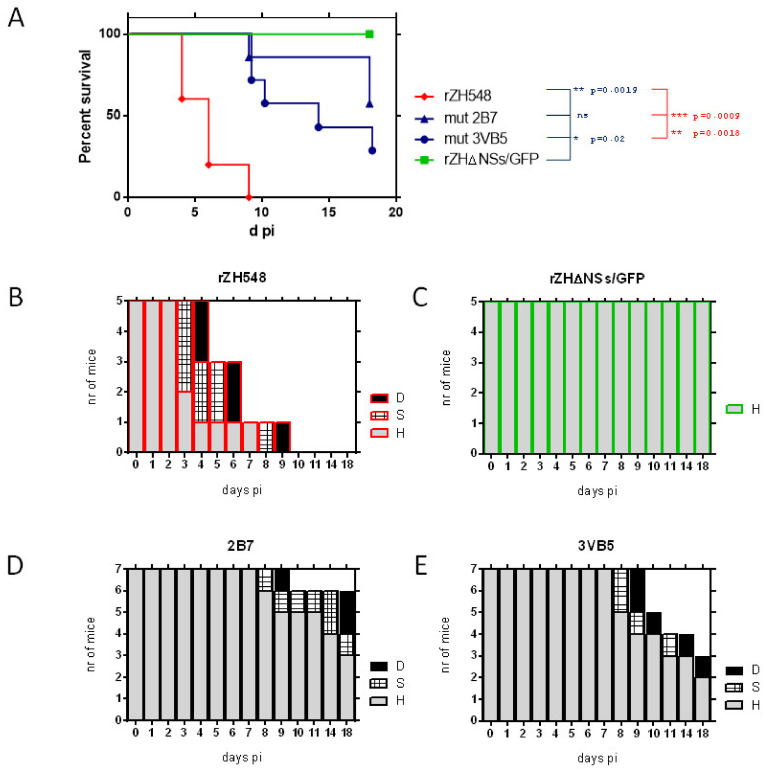
Analysis of the in vivo pathogenicity of the rZH548-P82L mutant viruses in BALB/c mice. 9–18-week-old male mice (*n* = 5–7, equally distributed) were inoculated IP with 500 plaque-forming units (pfus) of the indicated viruses and both rZH548-P82L clones, 2B7 and 3VB5. Wild-type rZH548 (red) and rZH548ΔNSs::GFP (labeled as, rZHΔNSs/GFP, green) viruses were included as controls for virulence and attenuation, respectively. Animals were monitored up to 18 days. (**A**) Survival rates and (**B**–**E**) morbidity upon challenge with the indicated viruses. The graph represents the clinical status of each mouse: D (dead/euthanized): black bars; S (signs-sick), hatched bars; H (healthy), grey bars. The animal within the group rZH548ΔNSs::GFP euthanized at day 4 pi was excluded from the survival analysis.

**Figure 2 viruses-13-00542-f002:**
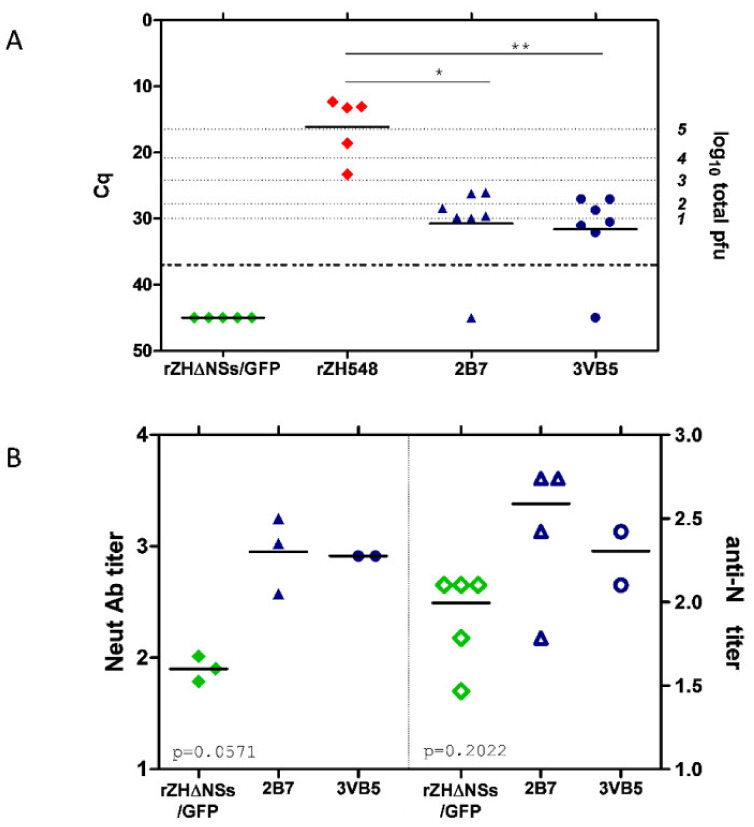
Viremia and seroconversion after inoculation with the rZH548-P82L mutant viruses. (**A**) Viremia. RT-qPCR on EDTA blood samples collected at day 3 pi. Samples giving a Cq (quantification cycle) value under the detection level of the assay (37) are arbitrarily represented as 45 and were excluded from the statistical analysis. The correlation of Cq data with pfu equivalents is indicated in the right Y axis. (**B**) Antibody responses in survivor mice at day 18 pi. Titers are expressed as the dilution of serum (log10) rendering a reduction in infectivity of 50% in a microneutralization assay (left Y-axis; closed symbols), and last dilution of serum (log10) giving an OD reading at 450 nm over 1.0 in anti-N ELISA (right Y-axis; open symbols). Each symbol corresponds to an individual mouse. For neutralization, only *n* = 3 samples were available for rZH548ΔNSs::GFP and 2B7. * *p* ≤ 0.05, ** *p* ≤ 0.001.

**Figure 3 viruses-13-00542-f003:**
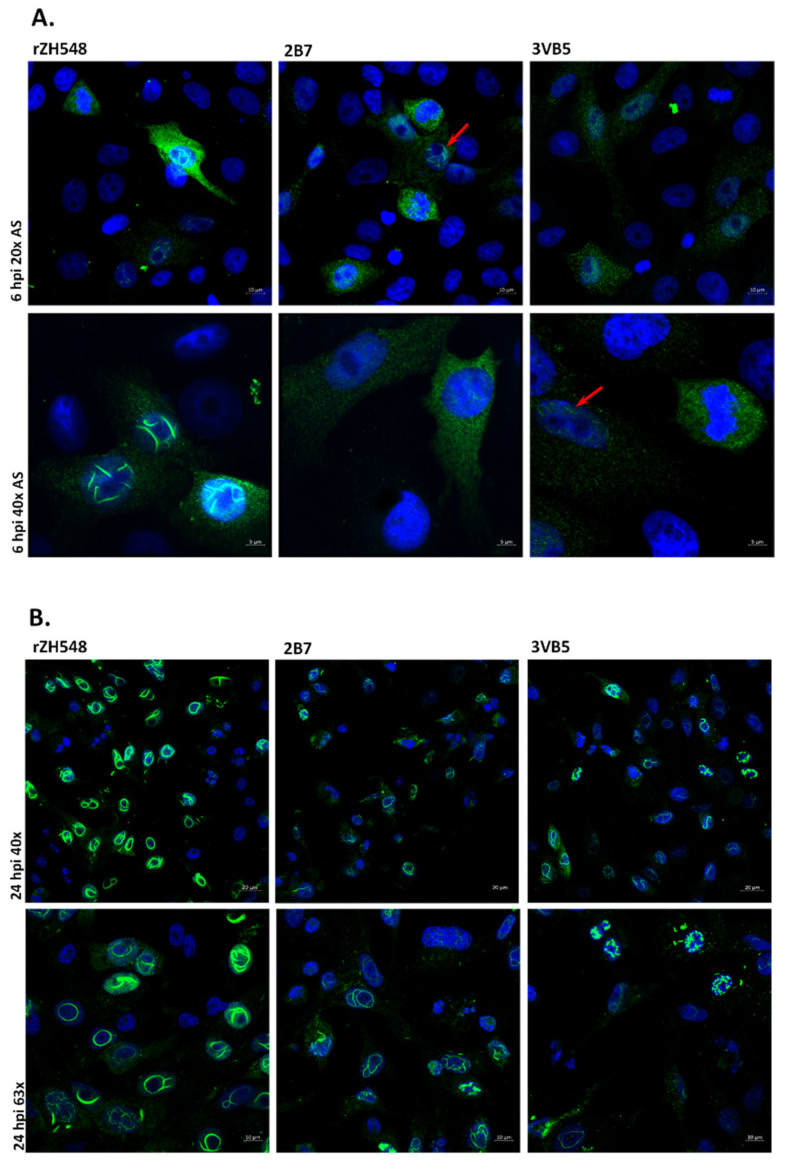
Localization and filament formation of wt and mutant NSs proteins. Vero cells were infected with rZH548 and the two rZH548-P82L mutants at a MOI of 1. At 6 (panel **A**) and 24 (panel **B**) hours pi, cells were fixed and subjected to indirect immunofluorescence with the anti-NSs monoclonal antibody 5C3A1B2. Nuclei were stained with DAPI. For each virus and time pi, 2 images with different magnification are shown as indicated. AS denotes Zeiss Airyscan 2D superresolution mode. Red arrows in panel **A** point to cells infected with the rZH548-P82L viruses where nuclear filaments could be detected. Scale bars: 20 µM (upper panel B), 10 µM (upper panel **A** and lower panel **B**); 5 µM (lower panel **A**).

**Figure 4 viruses-13-00542-f004:**
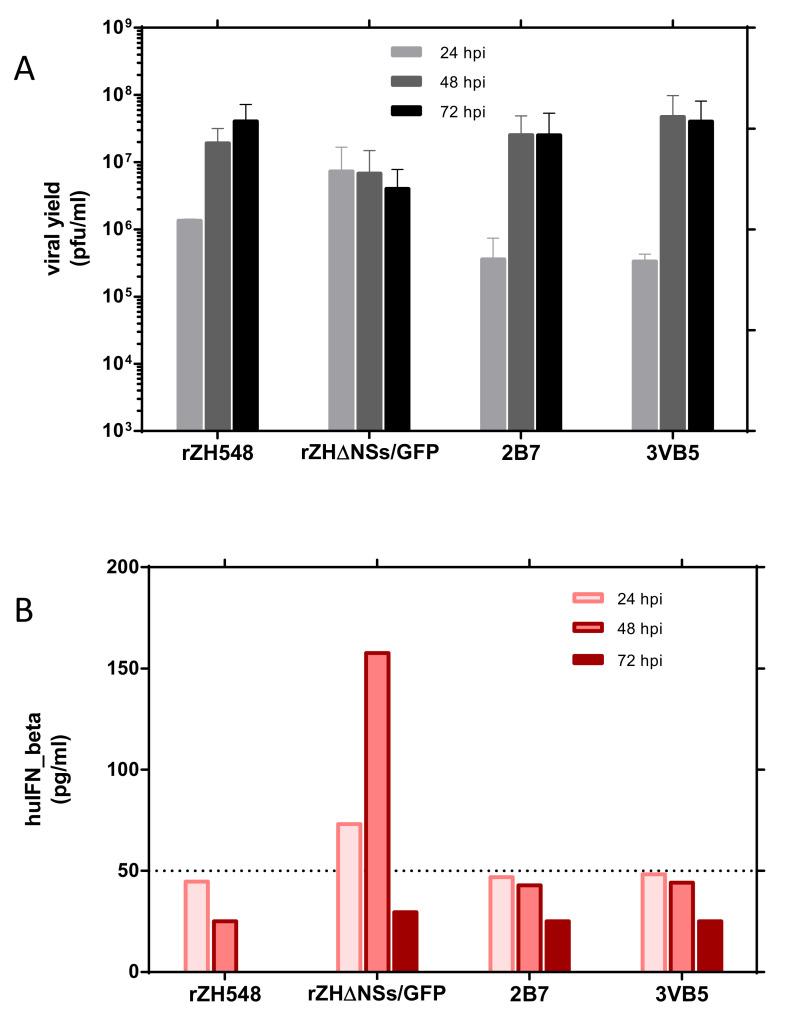
Growth of rZH548-P82L mutants on HEK293T cells and IFN-β production. HEK293T cells were infected at a MOI of 0.05 with the indicated viruses. At 24, 48 and 72 hpi, supernatants were collected and titrated on Vero cells (panel **A**) and analyzed for IFN-β production by ELISA (panel **B**). The limit of detection of this ELISA was established at 50 pg/mL (see Materials and Methods). The sample corresponding to rZH548 at 72 hpi was not analyzed in ELISA.

**Figure 5 viruses-13-00542-f005:**
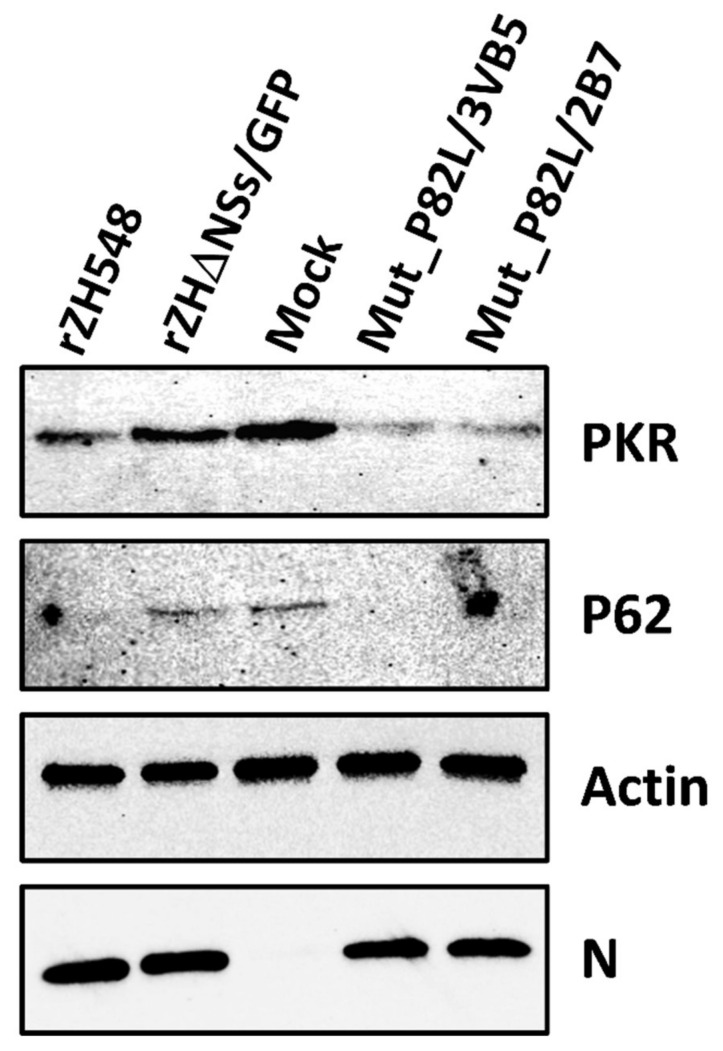
Degradation of protein kinase R (PKR) and p62 in rZH548-P82L infected cells. HEK293T cells were infected at a MOI of 1 with the indicated viruses. Cells were harvested at 20 hpi and analyzed by Western blot using anti-PKR (B-10), anti-p62 (H10) mouse monoclonal antibodies, anti RVFV-N mAb 2B1 and anti-actin antibody as primary antibodies. Samples loaded correspond to 10^6^ cells.

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
