# Peer review of "The Change P82L in the Rift Valley Fever Virus NSs Protein Confers Attenuation in Mice"

_viruses, 2021, doi:10.3390/v13040542_

Round 1

Reviewer 1 Report

The manuscript by Borrego et al., describes the introduction of an amino acid mutation in the NSs protein of Rift Valley fever virus and evaluation of the impact on virus propagation and virulence in mice.  The mutation evaluated here, P82L, was previously identified while evaluating a favipiravir escape mutant.  This mutation falls within a motif thought to be involved in virus inhibition of the type I IFN response.  The study described here found that viruses generated with the P82L mutations were slightly attenuated in Balb/c mice when compared to the wild-type ZH548 virus.  Further, the authors found a somewhat decreased viremia in infected mice and had a neutralizing Ab titer.  The authors proceeded to evaluate the potential impact of the P82L mutation by evaluating NSs intracellular localization in infected cells and the formation of NSs associated ‘nuclear filaments’. The authors argue that introduction of the P82L mutation prevents formation of these NSs ‘filaments’, but the data presented here are not particularly conclusive in my view.  Since NSs is a known type I IFN antagonist, the authors evaluated the virus yield and IFN B response in HEK 293 cells infected with ZH548 or the two mutant virus isolates.  Here they found essentially no difference between the WT virus and those with the P82L mutation although there was a large IFN-B response stimulated by a virus devoid of NSs.  Analysis of NSs induced intracellular signaling inhibition suggests that the P82L mutation leads to a degradation of PKR in infected cells, but that the mutation did not induce (or fail to inhibit) p62 expression as is seen in a virus devoid of NSs.

In general, this manuscript demonstrates that the P82L mutation in NSs of RVFV may play a role in virus induced pathogenesis in mice, and that a proline insertion at this site may impact virus induced inhibition of a type I IFN response.  However, the data presented here, outside of the mouse study, is fairly underwhelming.  I would like to have seen additional examination of the IFN regulatory pathway and a more convincing assessment of the ‘nuclear filament’ phenomenon.  There are a lot of questions left unanswered here, and I think additional work is warranted.

Specific comments:

Title:  I would delete the second part of the title, from “Not Related…”

Lines 112-13:  What was sequenced?  Was this the recovered virus and sequencing was done to confirm the mutations were retained?  Any compensatory mutations elsewhere in NSs?

Lines 195-199:  Was this animal tested for the presence of virus?  It seems odd to me that a submandibular bleed would induce tremors and a rapid 20% weight loss.  Also, the plot in figure 1A seems to indicate that the animal was included in the survival analysis, contrary to the statement on line 199 that it wasn’t.

Lines 206-210:  How is it that a survival rate of 2 from 7 is considered significant while a survival of 4 from 7 is not?  What is your comparator?  I suspect this is reversed.

Figure 3:  This figure is underwhelming.  The ‘filaments’ are barely visible and it isn’t clear what is being calculated in figure 3B.  I suggest including higher magnification images.  Further, the description for ‘filament’ analysis in the methods is not particularly clear.  Was the computer program allowed to determine what were/were not filaments or where the filaments mapped by hand?  I would be hesitant to allow a computer to do the analysis without some input from the user.  Also, the Figure 3 legend states that ‘two images’ were used in the analysis.  How many cells per field does that include?

Author Response

R1

The manuscript by Borrego et al., describes the introduction of an amino acid mutation in the NSs protein of Rift Valley fever virus and evaluation of the impact on virus propagation and virulence in mice.  The mutation evaluated here, P82L, was previously identified while evaluating a favipiravir escape mutant.  This mutation falls within a motif thought to be involved in virus inhibition of the type I IFN response.  The study described here found that viruses generated with the P82L mutations were slightly attenuated in Balb/c mice when compared to the wild-type ZH548 virus.  Further, the authors found a somewhat decreased viremia in infected mice and had a neutralizing Ab titer.  The authors proceeded to evaluate the potential impact of the P82L mutation by evaluating NSs intracellular localization in infected cells and the formation of NSs associated ‘nuclear filaments’. The authors argue that introduction of the P82L mutation prevents formation of these NSs ‘filaments’, but the data presented here are not particularly conclusive in my view.  Since NSs is a known type I IFN antagonist, the authors evaluated the virus yield and IFN B response in HEK 293 cells infected with ZH548 or the two mutant virus isolates.  Here they found essentially no difference between the WT virus and those with the P82L mutation although there was a large IFN-B response stimulated by a virus devoid of NSs.  Analysis of NSs induced intracellular signaling inhibition suggests that the P82L mutation leads to a degradation of PKR in infected cells, but that the mutation did not induce (or fail to inhibit) p62 expression as is seen in a virus devoid of NSs.

In general, this manuscript demonstrates that the P82L mutation in NSs of RVFV may play a role in virus induced pathogenesis in mice, and that a proline insertion at this site may impact virus induced inhibition of a type I IFN response.  However, the data presented here, outside of the mouse study, is fairly underwhelming.  I would like to have seen additional examination of the IFN regulatory pathway and a more convincing assessment of the ‘nuclear filament’ phenomenon.  There are a lot of questions left unanswered here, and I think additional work is warranted.

Response.- Thanks for the positive feedback. This work was intended as a first approach to understand the molecular basis of the reduced virulence observed in vivo. We thus focused on the prominent interferon antagonistic function of NSs (as this is a well-recognized virulence marker), by checking the levels of fundamental proteins involved in transcription/translation impairment (p62/PKR). Interestingly, the levels of both proteins were not affected and could not be linked to the attenuated phenotype. We fully agree that further investigation is needed to address if/how P82L mutation affects other NSs described roles (as those reviewed by Leventhal et al in this same special volume)

We also agree in the need for improving the nuclear filament images to better show the differences observed between wild-type and mutant viruses. Thus, in the revised version we have included improved figures to show these differences as well as more details to explain the quantification assessment performed.

Specific comments:

Title:  I would delete the second part of the title, from “Not Related…”

Response.-We have followed this suggestion in view of the general reviewers’ comments regarding the need for more in depth characterization of the IFN-signalling pathway.

Lines 112-13:  What was sequenced?  Was this the recovered virus and sequencing was done to confirm the mutations were retained?  Any compensatory mutations elsewhere in NSs?

We only sequenced to confirm the presence of the introduced mutation after virus rescue as well as for other compensatory mutations in the NSs ORF. We have added more explanatory sentences in M&M section. This has been also explained in the Results section.

Lines 195-199:  Was this animal tested for the presence of virus?  It seems odd to me that a submandibular bleed would induce tremors and a rapid 20% weight loss. Also, the plot in figure 1A seems to indicate that the animal was included in the survival analysis, contrary to the statement on line 199 that it wasn’t.

Response.-We have confirmed now that this mouse was not viremic. The strong signs observed in this animal after bleeding made us think that it was probably affected not by the bleeding itself but by an accidental brain damage during bleeding puncture or handling (signs were quite different from the ones expected in infected animals short after infection with RVFV). Therefore, we excluded it from formal statistical analysis. For a better understanding, we have now deleted it from the survival and morbidity plots (figures 1A and 1C) so that the new figures shows only a group of 5 animals with 100% of survival without clinical signs.

Lines 206-210:  How is it that a survival rate of 2 from 7 is considered significant while a survival of 4 from 7 is not?  What is your comparator?  I suspect this is reversed.

Response.-We are afraid that the text was confusing, so it has been rephrased. Survival time comparison between both P82L mutants was not significant (this has been now represented in the figure 1A). Stats data shown in figure 1A includes also comparisons of both mutants to the virulent strain (in red) or to the attenuated virus (in blue in the revised manuscript).

In addition, we also detected that the correct p value for both mutant comparison should be 0.1964. This has been now corrected in the revised script.

Figure 3:  This figure is underwhelming.  The ‘filaments’ are barely visible and it isn’t clear what is being calculated in figure 3B.  I suggest including higher magnification images.  Further, the description for ‘filament’ analysis in the methods is not particularly clear.  Was the computer program allowed to determine what were/were not filaments or where the filaments mapped by hand?  I would be hesitant to allow a computer to do the analysis without some input from the user.  Also, the Figure 3 legend states that ‘two images’ were used in the analysis.

Response.-We agree this should have been improved for clarity. We have included higher magnification images with more resolution in the script that we think allow to observe more clearly the differences between the wt and mutant viruses. Initially the use of a computer program was intended to ease the quantification of these differences. In this new version we have included quantification data form a visual inspection of different microscope fields and cell by cell counting and included the number of cells that were analyzed (Results heading 3.3, page 9).

Reviewer 2 Report

The manuscript by Borrego et al. describes an in vivo and in vitro investigation of a recombinant Rift Valley fever virus containing an amino acid substitution at position 82 of the non-structural protein of the small (S) segment, where a conserved proline (P) in a PXXP motif was changed to a leucine (L). This change is inspired by findings in an earlier study by the same group (still under consideration for peer-review publication, but available as a pre-print in bioRxiv archive) where a number of mutations were induced in the presence of an antiviral compound (favipiravir), resulting in a highly attenuated variant.

Borrego et al. studied two clones of the virus with the P82L change, on the backbone of a virulent wild-type strain, ZH548. This wild type was included in all analyses as a control, along with another control with an NSs deletion and presence of eGFP (attenuation control).

The authors show that both clones of the P82L recombinant viruses are attenuated (in vivo), although to a lesser extent than the control with NSs deletion and eGFP insert. To try and understand the attenuation mechanism, the authors did further in vitro studies to investigate NSs filament formation in infected cell nuclei and the effect on interferon beta.

The manuscript is well written and provides evidence of yet another attenuation marker that could potentially be targeted for the development of safe Rift valley fever live attenuated vaccines, particularly if combined with other single amino acid changes rather than large deletions in the NSs protein that might be poorly immunogenic. As the authors highlight in their concluding paragraph, more work is needed, though, to fully elucidate the mechanism of attenuation of P82L. 

The authors should consider the following minor comments when revising their manuscript:

Materials and Methods: the authors should include a separate sub-section describing the qRT-PCR that was used for analysis of viral loads, or alternatively just a more expansive description than what is currently included under section 2.4 Animal Inoculation and sampling. The authors cite one of their earlier papers for the qRT-PCR protocol, but it is unclear which of the primer sets used in that study, was used for this particular study. Also, it is unclear how qRT-PCR results were converted to pfu values presented in Figure 1. This should be more clearly described.

Sequencing: the authors chose to sequence only the NSs open reading frame to confirm the presence of the intended point mutation in the rescued virus clones, and in samples collected from the in vivo mouse trial. The results confirm that the mutation is maintained throughout, but based on the methodology this can only be said at the consensus level. Considering the ever decreasing cost of next generation sequencing technologies, and therefore the ability to sequence much deeper than just the consensus level, it might be important to investigate what is happening at the sub-consensus level. Minor variants at the sub-consensus level might also contribute to the pathogenicity of a virus stock. Although virus rescued from plasmids would be assumed to be more pure than stock prepared from a wild-type isolate, the stocks used in this study was passaged 3 times in BHK cells, which might have allowed generation of more diversity at the sub-consensus level. I am not suggesting that the authors should sequence deeper for the current study, but they might include a sentence or two to touch on this subject in the discussion.

Additionally, although mortality rates between the two clones of P82L were not different with statistical significance, there does appear to be a slight difference in how the mice responded to infection with the two clones. This might be attributable to normal variation between different experimental groups of relatively low numbers, but the authors did not sequence the L and M segments (and the rest of the S segment) to conclusively say the two clones were identical on a genome level, and that the induced mutation was the only difference between the recombinant viruses and the wild-type.

Figure 1A: the green lines and text on the right hand side which displays the statistical significance is difficult to read, particular in printed format. I suggest the authors change the colour to blue.

Figure 1A and C: One mouse that had apparent adverse reaction to the blood sampling on day 3 was removed from the study on day 4. The mouse was not included in the statistical analysis. Is this mouse still included in Figure 1A and C? Figure 1C shows morbidity for 6 mice in the rZHgfp group, while figure 2A show viremia results for 5 mice. It's unclear from section 2.4 which groups had 5, 6 or 7 animals at the start of the animal experiment. 

Author Response

R2

The manuscript by Borrego et al. describes an in vivo and in vitro investigation of a recombinant Rift Valley fever virus containing an amino acid substitution at position 82 of the non-structural protein of the small (S) segment, where a conserved proline (P) in a PXXP motif was changed to a leucine (L). This change is inspired by findings in an earlier study by the same group (still under consideration for peer-review publication, but available as a pre-print in bioRxiv archive) where a number of mutations were induced in the presence of an antiviral compound (favipiravir), resulting in a highly attenuated variant.

Borrego et al. studied two clones of the virus with the P82L change, on the backbone of a virulent wild-type strain, ZH548. This wild type was included in all analyses as a control, along with another control with an NSs deletion and presence of eGFP (attenuation control).

The authors show that both clones of the P82L recombinant viruses are attenuated (in vivo), although to a lesser extent than the control with NSs deletion and eGFP insert. To try and understand the attenuation mechanism, the authors did further in vitro studies to investigate NSs filament formation in infected cell nuclei and the effect on interferon beta.

The manuscript is well written and provides evidence of yet another attenuation marker that could potentially be targeted for the development of safe Rift valley fever live attenuated vaccines, particularly if combined with other single amino acid changes rather than large deletions in the NSs protein that might be poorly immunogenic. As the authors highlight in their concluding paragraph, more work is needed, though, to fully elucidate the mechanism of attenuation of P82L. 

The authors should consider the following minor comments when revising their manuscript:

Materials and Methods: the authors should include a separate sub-section describing the qRT-PCR that was used for analysis of viral loads, or alternatively just a more expansive description than what is currently included under section 2.4 Animal Inoculation and sampling. The authors cite one of their earlier papers for the qRT-PCR protocol, but it is unclear which of the primer sets used in that study, was used for this particular study. Also, it is unclear how qRT-PCR results were converted to pfu values presented in Figure 1. This should be more clearly described.

Response.-We thank very much the reviewer suggestion. A new paragraph has been introduced in M&M section (page 4) to clarify and to better describe the qRT-PCR methodology.

Sequencing: the authors chose to sequence only the NSs open reading frame to confirm the presence of the intended point mutation in the rescued virus clones, and in samples collected from the in vivo mouse trial. The results confirm that the mutation is maintained throughout, but based on the methodology this can only be said at the consensus level. Considering the ever decreasing cost of next generation sequencing technologies, and therefore the ability to sequence much deeper than just the consensus level, it might be important to investigate what is happening at the sub-consensus level. Minor variants at the sub-consensus level might also contribute to the pathogenicity of a virus stock. Although virus rescued from plasmids would be assumed to be more pure than stock prepared from a wild-type isolate, the stocks used in this study was passaged 3 times in BHK cells, which might have allowed generation of more diversity at the sub-consensus level. I am not suggesting that the authors should sequence deeper for the current study, but they might include a sentence or two to touch on this subject in the discussion. Additionally, although mortality rates between the two clones of P82L were not different with statistical significance, there does appear to be a slight difference in how the mice responded to infection with the two clones. This might be attributable to normal variation between different experimental groups of relatively low numbers, but the authors did not sequence the L and M segments (and the rest of the S segment) to conclusively say the two clones were identical on a genome level, and that the induced mutation was the only difference between the recombinant viruses and the wild-type.

Response.-We fully agree with this observation.  We are aware of the limitations of our sequencing data consisting only of a consensus sequence for the NSs gene and no other data on the rest of the viral genome. We have included this consideration in the discussion (p12-13) since it is true that sub consensus viral variants may modulate virulence/pathogenicity. In fact, the rationale for assaying two independent clones was to strength the role of this change, provided other genomic variants could arise upon virus replication.

We agree that in the absence of the whole sequences we cannot conclusively say that the two S279-mutant viruses 2B7 and 3VB5 are identical and that the induced mutation is the only difference between them and the wild-type. In our opinion this indeed emphasizes the attenuating role of the P82L change, since for both clones other genetic differences not identified may exist and account for the apparent differences found. Moreover, a third rescued P82L mutant clone (not included in the study) confirmed the attenuated phenotype in mice.

Our sequencing data were conclusive and no mixed sequences were observed in the corresponding mutated position. Thus, we decided that this approach of consensus sequencing was good enough to determine (i) that the introduced mutation was maintained in the viruses to be inoculated and that no other (compensatory?) changes were introduced in the NSs sequence; and (ii) the stability of this substitution in an in vivo system in samples collected 3 dpi from infected animals.

On the other hand, some works have reported about the variability generated in recombinant viruses rescued by a reverse genetics system after a short number of cell passages (Lokugamage et al, J. of Virology 2012; Ikegami, J. Virol 2021). Based on these data, we decided not to check the whole genome of our recombinant viruses assuming that our inocula would display mostly the original sequence, with only minor variants, but the generation and imposition of a new variant was considered to be an extremely improbable event.

Figure 1A: the green lines and text on the right hand side which displays the statistical significance is difficult to read, particular in printed format. I suggest the authors change the colour to blue.

We thank for the suggestion. This has been changed as suggested.

Figure 1A and C: One mouse that had apparent adverse reaction to the blood sampling on day 3 was removed from the study on day 4. The mouse was not included in the statistical analysis. Is this mouse still included in Figure 1A and C? Figure 1C shows morbidity for 6 mice in the rZHgfp group, while figure 2A show viremia results for 5 mice. It's unclear from section 2.4 which groups had 5, 6 or 7 animals at the start of the animal experiment. 

We have confirmed now that this mouse was not viremic. Therefore, we excluded it from formal statistical analysis. For a better understanding, we have now deleted it from the survival and morbidity plots (figure 1A and 1C) so that the new figures shows only a group of 5 animals with 100% of survival without signs.

Reviewer 3 Report

In this manuscript authors are building on a previous study in which they generated a favipiravir-resistant RVFV strain containing 24 AA changes that was attenuated in vivo. Here the authors reverse-engineered one of those mutations, P82L, which is located in the RVFV virulence factor NSs, and investigate its effect. Recombinant RVFV viruses containing this mutation are less lethal in mice than the wt virus. Since RVFV NSs is a IFN I antagonist, authors speculated initially that this mutation resulted in an attenuated phenotype by affecting this antagonistic function, but conclude that P82L results in attenuation via a mechanism unrelated to type I IFN antagonism.

Major comments

  1. My first main concern with this manuscript centers around the fact that the authors chose to use two separate clones containing the NSs P82L mutation for this study. The rationale for this is unclear, assuming they are identical and only contain the P82L mutation? These clones certainly do not behave identically in vivo. It appears that the authors only sequenced the NSs gene and not the rest of the virus, where e.g. compensatory mutations could have arisen. The fact that there doesn’t appear to be a link to previously established functions of NSs (IFN I suppression) suggests that the differences in vivo might be due to a location outside of NSs. The whole viral genome should be sequenced, preferably by NGS.
  2. My second main concern relates to the authors’ conclusion that P82L does not affect NSs IFN I antagonistic functions. While they do present some evidence supporting this, it can be strengthened. For example, based on the authors’ previous publication (Frontiers 2021), it is unclear why authors didn’t also analyze growth kinetics in C6/36 cells and Vero cells, since in these cell lines different growth kinetics could be observed for the mutant virus containing 24 AA changes. Furthermore, RVFV likely encodes more immunomodulatory proteins, which can mask the effect of NSs-P82L. Performing assays using only the overexpressed NSs-P82L could further strengthen these findings, since other viral immunomodulatory proteins are absent in those settings. In the absence of additional supporting information that P82L does not affect NSs type I IFN antagonistic activity, these claims should be moderated.
  3. Besides the observed in vivo attenuation, the effect of P82L on NSs nuclear localization and filament formation is the most interesting finding. However, the data presented in Figure 3 is not of sufficient quality to support the claims made by the authors. The cells should be shown at a higher magnification. It also appears that only a low number of imaged cells are infected. Please indicate how many infected cells were analyzed to determine filament size, not how many images. With the low number of infected cells per image, this appears to be based on only a handful of filaments. Furthermore, a size bar needs to be included in the figure.
  4. Since on the last study day (day 18) animals still died/had to be euthanized, the study was not designed long enough. The animals should have been followed for a longer time period.
  5. It appears that the original experiment generating the favipiravir-resistant mutant was carried out using the 56/74 strain, whereas in this study the P82L mutation was added to the ZH584 backbone. How similar are these two strains?
  6. The Material and Methods section requires some more details for most assays. It is not sufficient to refer to other publications (e.g. viral infections, viral RNA extractions, sequencing, development of disease in mice, neutralization assay, western blot, plaque assays).
  7. Fig 4: It is unclear why the IFN-beta ELISA data is included in the same figure as the viral titer/growth kinetics. It would make more sense to show that data in a separate panel. Why does the ELISA data not have standard deviations? Was only a single well tested? Line 287: “IFN-β was only detected in samples recovered 48 hpi from cells infected with rZH548ΔNSs::GFP.” - That is not what figure 4 shows. All 24hpi values seem well above the limit of detection. Also please indicate that the line indicates the LOD, either in the figure or in the figure legend.

Minor comments

  1. Line 20: “These results unveil a new RVFV virulence marker highlighting the multiple ways of NSs protein to modulate viral infectivity” – It is unclear how the presented data ‘unveils a new RVFV virulence marker’. RVFV NSs is already described as a virulence marker. Authors didn’t provide sufficient evidence that this mutation did not affect IFN I signaling.
  2. Line 91: Please provide the catalog # of the IFN-beta ELISA kit used
  3. Line 164: “After checking the correct sequence of the resulting plasmid, co-cultures of HEK293 and BHK-21 cells were transfected in triplicates.” – please indicate that this is a rescue attempt, this sentence implies that just the mutant S plasmid was transfected.
  4. Methods section and line 168: (..) “screen for the presence of mutant virus by the appearance of cytopathic effect (CPE)”. – CPE can indicate the presence of rescued virus, but it won’t be able to tell you if it is the mutant virus, or inadvertently the wt virus.
  5. In most figures, rZH548ΔNSs::GFP is abbreviated to rZHgfp. This suggests that it’s a GFP-encoding reporter version of the wild-type virus, whereas the emphasis should be on the fact that it doesn’t contain NSs. Please change rZHgfp to rZHΔNSs/GFP or something similar.
  6. Fig 1A: Why is there a relatively large age difference (9-16/18 weeks) between the mice? Where these animals distributed equally over all groups? Was there a difference in observed pathogenicity? The Methods section states 9-18 weeks, whereas the figure legend lists 9-16 weeks.
  7. Fig 1B-D: The text states that these animals were monitored daily, however that is not reflected in the figure past day 11
  8. Authors refer to the data depicted in Figure 1 as ‘infectivity’. However, ‘pathogenicity’ would be more accurate.
  9. Line 213: ‘immunized’ is not the most accurate description here
  10. Line 216: “randomly selected samples” – please be more specific. Blood samples? Distributed how?
  11. Fig 2A: how do authors correlate Cq values to log virus titer? Did they use a standard curve in parallel? This should be described in the Methods section.
  12. Line 229/Fig 2A: “Samples giving a Cq value under the detection level of the assay, 37, are arbitrarily represented as 45 and were excluded from the statistical analysis.” – There appears to be a dot at ~38 (below LOD) for 3VB5.
  13. Figure 4: please change the in-figure legend to something a bit more intuitive (there is no real advantage of abbreviating ‘viral yield’ for the later time points, same for indicating that it is human IFN).
  14. Fig 5: please replace 1-5 with the virus mutants’ names
  15. Line 317: “This mutation was originally identified in a virus isolated under the selective pressure of a mutagenic agent and found later to be hyper-attenuated in mice” – the way the sentence is structured now suggests that this single mutation was responsible for the hyper-attenuation in vivo in the previous paper
  16. It’s a bit confusing that authors flip back and forth between S279 and P82L mutant viruses
  17. In various places authors use HEK293. It is unclear if HEK293T is meant. If not, HEK293 cells should be included in the ‘Cells’ section of the Material and Methods

Author Response

R3

In this manuscript authors are building on a previous study in which they generated a favipiravir-resistant RVFV strain containing 24 AA changes that was attenuated in vivo. Here the authors reverse-engineered one of those mutations, P82L, which is located in the RVFV virulence factor NSs, and investigate its effect. Recombinant RVFV viruses containing this mutation are less lethal in mice than the wt virus. Since RVFV NSs is a IFN I antagonist, authors speculated initially that this mutation resulted in an attenuated phenotype by affecting this antagonistic function, but conclude that P82L results in attenuation via a mechanism unrelated to type I IFN antagonism.

Major comments

  1. My first main concern with this manuscript centers around the fact that the authors chose to use two separate clones containing the NSs P82L mutation for this study. The rationale for this is unclear, assuming they are identical and only contain the P82L mutation? These clones certainly do not behave identically in vivo. It appears that the authors only sequenced the NSs gene and not the rest of the virus, where e.g. compensatory mutations could have arisen. The fact that there doesn’t appear to be a link to previously established functions of NSs (IFN I suppression) suggests that the differences in vivo might be due to a location outside of NSs. The whole viral genome should be sequenced, preferably by NGS.

A similar comment has been arisen by Reviewer 2. Based on published data about the variability generated in recombinant viruses rescued by a reverse genetics system after a short number of cell passages (Lokugamage et al, J. of Virology 2012; Ikegami, J. Virol 2021), we decided to sequence the corresponding ORF (NSs) to confirm the mutation was maintained and to check that no other changes had been introduced in this protein (at least those that could be detected in a consensus sequence). In the absence of sequence data from the rest of the viral genome, we assumed that our inocula would display mostly as the original sequence, with low diversity, and considered that the generation and imposition of a new variant would be an improbable event. This low variability was also assumed for the rZH548 rescued wild-type virus, obtained by the same procedure, with the same set of plasmids and subjected to the same cycles of growing in cell culture.

Keeping this in mind, we decided to assay 2 different clones where only the consensus sequence of NSs had been checked, as a kind of replica that could act as control of any effect due to any other potential changes introduced all along the genome because of this variation. Furthermore, a third clone was also tested in the mice assays with equivalent results of attenuation and so was not included in the following cell experiments nor in the paper.

While results in all the experiments performed in cell culture were indistinguishable, we agree that there does appear to be a slight difference in how the mice responded to infection with the two clones, although mortality rates were not different with statistical significance. We don’t know whether this might be attributable to normal variation between experimental groups of low numbers of individuals, or to genetic differences not identified (at the subconsensus level for the NSs and/or in the rest of the genome). In any case we think that our results prove solidly the attenuating effect of the mutation P82L.

In our opinion, the fact that there doesn’t appear to be a link to the previously established functions of NSs as IFN I suppressor suggests that the differences in vivo might be due to any other function still to be determined, whose elucidation goes beyond the goals of this work.

  1. My second main concern relates to the authors’ conclusion that P82L does not affect NSs IFN I antagonistic functions. While they do present some evidence supporting this, it can be strengthened. For example, based on the authors’ previous publication (Frontiers 2021), it is unclear why authors didn’t also analyze growth kinetics in C6/36 cells and Vero cells, since in these cell lines different growth kinetics could be observed for the mutant virus containing 24 AA changes. Furthermore, RVFV likely encodes more immunomodulatory proteins, which can mask the effect of NSs-P82L. Performing assays using only the overexpressed NSs-P82L could further strengthen these findings, since other viral immunomodulatory proteins are absent in those settings. In the absence of additional supporting information that P82L does not affect NSs type I IFN antagonistic activity, these claims should be moderated.

We agree that our analysis on the NSs functions is limited and there is room for further investigations, since this protein has been described to function through a variety of mechanisms and impact a number of cell processes.  We thank the reviewer for the suggestions of experimental approaches aimed to strengthen our findings, and in the absence of new supporting information, we agree to moderate our conclusions. So the second part of the title has been deleted.

  1. Besides the observed in vivo attenuation, the effect of P82L on NSs nuclear localization and filament formation is the most interesting finding. However, the data presented in Figure 3 is not of sufficient quality to support the claims made by the authors. The cells should be shown at a higher magnification. It also appears that only a low number of imaged cells are infected. Please indicate how many infected cells were analyzed to determine filament size, not how many images. With the low number of infected cells per image, this appears to be based on only a handful of filaments. Furthermore, a size bar needs to be included in the figure

We have improved the quality of the confocal images to better show the differences found. We also have indicated the number of infected cells analyzed to demonstrate such differences.

  1. Since on the last study day (day 18) animals still died/had to be euthanized, the study was not designed long enough. The animals should have been followed for a longer time period.

The reviewer is right that if the experiment should have been followed for a longer time, final numbers of survival would be probably different, probably allowing also a more accurate conclusion about whether differences between the 2 mutant clones tested were significant or not. Even though this would have provided quite interesting information, we think that the main conclusion of our work, i.e. the attenuation provided by the change introduced, is still solid. Results shown here, with a 18-day experiment already demonstrate a statistical significant difference in disease development and survival when comparing the mutants to the parental rZH548, with mean survival times of 6 days (for rZH548), 14 days (for mutant clone 3VB5) and undefined for mutant clone 2B7.

  1. It appears that the original experiment generating the favipiravir-resistant mutant was carried out using the 56/74 strain, whereas in this study the P82L mutation was added to the ZH584 backbone. How similar are these two strains?

It is broadly accepted that RVFV diversity is relatively low, with identity differences between different strains of only approximately 5% and 2% at the nucleotide and amino acid levels, respectively (Bird et al , J.Virol 2007). In this paper seven different lineages could be determined. We stated in the discussion section the RVFV strain 56/74 derived from a South African origin classified into lineage D, while the strain ZH548 belongs to lineage A . Our own sequencing data, to be soon published on Genbank, shows a difference between both viral strains of 3.7% (237/6404; segment L), 2.9% (114/3885; segment M) and 3.7% (63/1691; segment S), at the nucleotide level.

  1. The Material and Methods section requires some more details for most assays. It is not sufficient to refer to other publications (e.g. viral infections, viral RNA extractions, sequencing, development of disease in mice, neutralization assay, western blot, plaque assays).

Following reviewer’s comments we have now added more details and additional explanations considered more specific for the RVFV subject.

  1. Fig 4: It is unclear why the IFN-beta ELISA data is included in the same figure as the viral titer/growth kinetics. It would make more sense to show that data in a separate panel. Why does the ELISA data not have standard deviations? Was only a single well tested? Line 287: “IFN-β was only detected in samples recovered 48 hpi from cells infected with rZH548ΔNSs::GFP.” - That is not what figure 4 shows. All 24hpi values seem well above the limit of detection. Also please indicate that the line indicates the LOD, either in the figure or in the figure legend.

Since samples analysed for huIFN beta were the same supernatants whose titers are shown in the figure, we thought that it could be helpful to show these results in the same figure. Now, following the reviewer suggestions, the figure 4 has been split in two panels.

Although each sample was tested in duplicates, we represented these ELISA data without standard deviations because of our procedure: each sample rendered 2 OD readings used to obtained a mean OD value, and this mean OD value (one single value) was the one interpolated in the standard curve to get the IFN concentration. This has been explained in the Methods section

The sentence in line 287 refers to samples that were considered negative after visual examination of the ELISA plates and with OD readings close to the blank, even though calculations run out for each and every sample rendered a numerical value. Now all the raw data (available for the reviewer) have been revised and recalculated according to the procedure described in the new paragraph added to M&M and a new figure (figure 4 panel B, including also data obtained with 72 hpi samples) has been represented.

Minor comments

  1. Line 20: “These results unveil a new RVFV virulence marker highlighting the multiple ways of NSs protein to modulate viral infectivity” – It is unclear how the presented data ‘unveils a new RVFV virulence marker’. RVFV NSs is already described as a virulence marker. Authors didn’t provide sufficient evidence that this mutation did not affect IFN I signaling.

We have changed the sentence in the abstract: “These results highlight the key role of NSs protein in modulation of viral infectivity”

  1. Line 91: Please provide the catalog # of the IFN-beta ELISA kit used

This ahas been indicated as requested. Besides, in order to clarify some questions regarding figure 4 (point 7 of this reviewer) a new paragraph has been introduced in M&M section describing this procedure in detail. 

  1. Line 164: “After checking the correct sequence of the resulting plasmid, co-cultures of HEK293 and BHK-21 cells were transfected in triplicates.” – please indicate that this is a rescue attempt, this sentence implies that just the mutant S plasmid was transfected.

Corrected as suggested. The sentence says: “co-cultures of HEK293 and BHK-21 cells were transfected in triplicates with the whole set of plasmids constituting our reverse genetic system.”

  1. Methods section and line 168: (..) “screen for the presence of mutant virus by the appearance of cytopathic effect (CPE)”. – CPE can indicate the presence of rescued virus, but it won’t be able to tell you if it is the mutant virus, or inadvertently the wt virus.

Corrected, the word mutant has been deleted

  1. In most figures, rZH548ΔNSs::GFP is abbreviated to rZHgfp. This suggests that it’s a GFP-encoding reporter version of the wild-type virus, whereas the emphasis should be on the fact that it doesn’t contain NSs. Please change rZHgfp to rZHΔNSs/GFP or something similar.

Corrected as suggested

  1. Fig 1A: Why is there a relatively large age difference (9-16/18 weeks) between the mice? Where these animals distributed equally over all groups? Was there a difference in observed pathogenicity? The Methods section states 9-18 weeks, whereas the figure legend lists 9-16 weeks.

The corrected age range is 9-18 weeks- changed in the figure

The age of mice in this work was conditioned by both the availability of mice and the schedule allowed for use of the high containment animal facilities. Even though this age difference, that could correspond to different life phases (young adults 8-12 week-old; mature adults 3-6 month-old) we decided to go on with the experiment as planned, and distributed the animals so that each group included representatives of the different ages. No differences in pathogenicity correlating to the age were observed.

  1. Fig 1B-D: The text states that these animals were monitored daily, however that is not reflected in the figure past day 11

The reviewer is right. Daily has been deleted from the text both in the figure legend and in M&M.

  1. Authors refer to the data depicted in Figure 1 as ‘infectivity’. However, ‘pathogenicity’ would be more accurate.

Corrected as suggested

  1. Line 213: ‘immunized’ is not the most accurate description here

Agree. We have changed immunized by either “infected” or “inoculated”

  1. Line 216: “randomly selected samples” – please be more specific. Blood samples? Distributed how?

The following sentence has been introduced for clarity:

“A total of 6 RNA samples (3 from each mutant group, randomly selected) extracted from day 3pi blood were used for RT-PCR amplification and NSs sequencing.

  1. Fig 2A: how do authors correlate Cq values to log virus titer? Did they use a standard curve in parallel? This should be described in the Methods section.

A new paragraph has been introduced in M&M section to clarify this point, and a sentence in the corresponding figure legend.

  1. Line 229/Fig 2A: “Samples giving a Cq value under the detection level of the assay, 37, are arbitrarily represented as 45 and were excluded from the statistical analysis.” – There appears to be a dot at ~38 (below LOD) for 3VB5.

The reviewer is right. This dot corresponds to a sample rendering a value of 37.7 that should have been considered as negative, and represented as 45. This has been now corrected in a new figure and the corresponding statistical analysis recalculated.

  1. Figure 4: please change the in-figure legend to something a bit more intuitive (there is no real advantage of abbreviating ‘viral yield’ for the later time points, same for indicating that it is human IFN).

This has been corrected as suggested

  1. Fig 5: please replace 1-5 with the virus mutants’ names

This has been corrected as suggested

  1. Line 317: “This mutation was originally identified in a virus isolated under the selective pressure of a mutagenic agent and found later to be hyper-attenuated in mice” – the way the sentence is structured now suggests that this single mutation was responsible for the hyper-attenuation in vivo in the previous paper

 We have changed the sentence to avoid misinterpretation: this mutation was one of the 24 amino acid changes originally identified in a virus isolated under the selective pressure of a mutagenic agent that found to be hyper-attenuated in mice” 

  1. It’s a bit confusing that authors flip back and forth between S279 and P82L mutant viruses

We have now reviewed the text following this indication and correct this trying to avoid any confusion. The viruses are still named after the genomic change S279, but in some places their names have been shortened so that they are referred just as “mutant viruses”.  “P82L” has been deleted or substituted in many sentences (“change P82L” by “aa change introduced” or equivalent); besides, additional clarifying sentences/paragraphs have been included.

  1. In various places authors use HEK293. It is unclear if HEK293T is meant. If not, HEK293 cells should be included in the ‘Cells’ section of the Material and Methods

We used HEK293T cells. This has been corrected along the manuscript. 

Round 2

Reviewer 1 Report

The authors have made considerable improvements to this manuscript and have addressed my primary concerns.  I do ask though that the authors indicate in the paragraph beginning at line 256 that the control animal discussed here was tested for virus and found to be negative.  The response to reviewers indicated that this was done, best to include in the manuscript.

Author Response

The authors have made considerable improvements to this manuscript and have addressed my primary concerns.  I do ask though that the authors indicate in the paragraph beginning at line 256 that the control animal discussed here was tested for virus and found to be negative.  The response to reviewers indicated that this was done, best to include in the manuscript.

We thank the reviewer for his/her kind comments. We have included this suggestion, indicating that “upon RT-qPCR testing it was found virus negative.”

Reviewer 3 Report

  • I would have preferred that the authors had taken the effort to sequence the whole viral genome of their mutants. Their argument that little changes have been observed for rescued wild type virus might not hold true for these mutant viruses, in which additional pressure to acquire compensatory mutations is present. In the absence of additional sequencing, authors should consider including the in vivo data that they have of the 3rd P82L isolate. In my opinion this would strengthen the paper, even if its just the survival curve.
  • The expanded MM section is a vast improvement; however, some details are still missing:
    • What kind of cells and media was used in the plaque assay?
    • What kind of tubes were used to collect blood for RNA extraction and sera collection? I think this is mentioned somewhere in a figure legend, but it would be useful to add it to the MM as well.
    • Could the authors explain the contents or manufacturer of the “SDS-PAGE sample buffer”?
  • By my comment that using both S279 and P82L interchangeably was a little confusing, I did not mean taking P82L out in many places. I think P82L is more intuitive, and it is used more throughout the manuscript, including the title. I assume the ‘S’ refers to the S segment, which makes it unnecessary complicated, as most people would assume it stands for ‘serine’ in this context. Using the ‘S’ to indicate S segment would be more logical if also mutations in other segments were discussed.

Author Response

  • I would have preferred that the authors had taken the effort to sequence the whole viral genome of their mutants. Their argument that little changes have been observed for rescued wild type virus might not hold true for these mutant viruses, in which additional pressure to acquire compensatory mutations is present. In the absence of additional sequencing, authors should consider including the in vivo data that they have of the 3rdP82L isolate. In my opinion this would strengthen the paper, even if its just the survival curve.

We totally agree this would be an interesting study that, unfortunately at this moment, we cannot afford, but definitively we should undertake a soon as we have more resources available. As suggested, we have included available data on survival of an additionally rescued virus clone (named 3B5). We have included both survival curves and clinical scores in a supplemental figure. See in Discussion section page 12. We agree this will help to strength our conclusions.

  • The expanded MM section is a vast improvement; however, some details are still missing:
      • What kind of cells and media was used in the plaque assay?
      • What kind of tubes were used to collect blood for RNA extraction and sera collection? I think this is mentioned somewhere in a figure legend, but it would be useful to add it to the MM as well.
      • Could the authors explain the contents or manufacturer of the “SDS-PAGE sample buffer”?

We´d like to thank the reviewer for helping us to improve the quality of the paper. All these missing details have now been included in the new revised version.

  • By my comment that using both S279 and P82L interchangeably was a little confusing, I did not mean taking P82L out in many places. I think P82L is more intuitive, and it is used more throughout the manuscript, including the title. I assume the ‘S’ refers to the S segment, which makes it unnecessary complicated, as most people would assume it stands for ‘serine’ in this context. Using the ‘S’ to indicate S segment would be more logical if also mutations in other segments were discussed.

Clearly, we misinterpreted this comment. We have now revised the changes made and returned to the “P82L” term instead of “S279”. The reviewer is right and we used “S” to refer changes found in the S-segment (as opposed to other mutations found in either M or L segment, in our previous paper). For the sake of clarity, we have shifted from “S279 mutant” to “P82L mutant”.